# ROP16 of *Toxoplasma gondii* Inhibits Innate Immunity by Triggering cGAS-STING Pathway Inactivity through the Polyubiquitination of STING

**DOI:** 10.3390/cells12141862

**Published:** 2023-07-15

**Authors:** Qi-Wang Jin, Ting Yu, Ming Pan, Yi-Min Fan, Si-Yang Huang

**Affiliations:** 1Jiangsu Key Laboratory of Zoonosis, Jiangsu Co-Innovation Center for Prevention and Control of Important Animal Infectious Diseases and Zoonosis, Institute of Comparative Medicine, College of Veterinary Medicine, Yangzhou University, Yangzhou 225009, China; 2Joint International Research Laboratory of Agriculture and Agri-Product Safety, The Ministry of Education of China, Yangzhou University, Yangzhou 225009, China

**Keywords:** *Toxoplasma gondii*, ROP16, cGAS-STING pathway, immune escape

## Abstract

cGAS-STING signaling is a major pathway in inducing type Ⅰ IFN, which plays a crucial role in the defense against *T. gondii* infection. In contrast, *T. gondii* develops multiple strategies to counteract the host defense, causing serious diseases in a wide range of hosts. Here, we demonstrate that *T. gondii* rhoptry protein 16 (ROP16) dampens type I interferon signaling via the inhibition of the cGAS (cyclic GMP-AMP synthase) pathway through the polyubiquitination of STING. Mechanistically, ROP16 interacts with STING through the SignalP domain and inhibits the K63-linked ubiquitination of STING in an NLS (nuclear localization signal)-domain-dependent manner. Consequently, knocking out the ROP16 in PRU tachyzoites promotes the STING-mediated production of type I IFNs and limits the replication of *T. gondii.* Together, these findings describe a distinct pathway where *T. gondii* exploits the ubiquitination of STING to evade host anti-parasite immunity, revealing new insights into the interaction between the host and parasites.

## 1. Introduction

*Toxoplasma gondii* is an obligate intracellular protozoan that is widely disseminated among humans and other warm-blooded animals [1,2]. *T. gondii* is an opportunistic pathogen, which is usually asymptomatic in immune-competent individuals but causes serious toxoplasmosis in immune-suppressive patients [1]. Based on the clinical symptoms, *T. gondii* infection can be divided into two stages: acute and stable chronic. During the acute stage, *T. gondii* tachyzoite rapidly replicates in parasitophorous vacuoles (PVs), while during the chronic stage, the tachyzoite transitions into a bradyzoite, and the cysts reside within the tissue [3]. The PV membrane (PVM) and cyst wall protect *T. gondii* from host immune responses. Once they have invaded the host cells, various pathogen-associated molecular patterns (PAMPs), including DNA and proteins, are released into the host cytoplasm through PVM and the cyst membrane [4,5,6,7]. The host innate immune system has evolved to recognize *T. gondii* PAMPs secreted by *T. gondii* through germline-encoded pattern recognition receptors (PRRs), such as Toll-like receptors (TLRs) and NOD-like receptors (NLRs) [4,8,9,10,11,12,13]. The innate immune signaling pathways have been extensively studied, especially regarding TLRs recognizing protein from *T. gondii*. Beyond these canonical PRRs, various cytoplasmic DNA receptors can also sense these PAMPs to induce the expression of type Ⅰ interferon.

Cyclic GMP-AMP synthase (cGAS) is a recently discovered cytoplasmic DNA sensor that regulates the immune response against various pathogens, such as viruses, bacteria, and parasites [14,15,16]. The signaling process begins with the binding of the pathogen or host DNA to cGAS in the cytoplasm. The activated cGAS utilizes ATP and GTP as substrates to generate cyclic GMP-AMP, a secondary messenger that binds to and activates the stimulator of IFN genes (STING). Activated STING recruits TANK-binding kinase 1 (TBK1), which undergoes trans-phosphorylation and further activates STING. The transcription factor IFN regulatory factor 3 (IRF3) is recruited to STING and further phosphorylated by activated TBK1. Phosphorylated IRF3 forms dimers and trans-locates into the nucleus to induce the expression of type Ⅰ IFNs and other inflammatory cytokines, which promote anti-infection immune responses. Additionally, activated STING can independently induce another transcription factor, nuclear factor-κB (NF-κB), in a TBK1-independent manner to enhance immune responses at the same time [16,17,18,19].

The cGAS-STING pathway is involved in *T. gondii* infection [20,21]. In a previous study, the authors found that cGAS- or STING-deficient mice were much more susceptible to a lethal infection of *T. gondii* than wild-type mice [20], while another research study mechanistically found that ROP18 from RH strains could suppress the type Ⅰ interferon by inhibiting cGAS-STING signaling [21]. However, the mechanism underlying the regulation of this pathway by *T. gondii* remains unclear. As an obligate intracellular protozoan, *T. gondii* secretes numerous proteins into the host cytoplasm, which can interact with the host immune system to regulate infection. In this study, the following question is of high interest: Are there any virulence factors that target the cGAS-STING pathway to regulate host immune responses? To answer this question, we screened cGAS-STING signaling regulators from the proteins secreted by *T. gondii*. Fortunately, we found that ROP16 from the PRU strain of *T. gondii* (ROP16) has the effect of inhibiting this pathway. 

ROP16, a putative protein kinase, is injected into host cells during *T. gondii* infection [22,23]. Many studies have indicated that ROP16 could translocate to the host nucleus and regulate immune responses by targeting the signal transducer and activator of the transcription (STAT) pathway [6,22,23,24,25,26,27]. However, little is known about the influence of ROP16 on cGAS-STING signaling. In this study, we found that ROP16 inhibited type Ⅰ IFN immune responses by regulating cGAS-STING signaling. Next, we found that ROP16 inhibited the activation of STING by suppressing the K63-linked polyubiquitination of STING. Furthermore, we found that ROP16 inhibited the cGAS-STING pathway in a predicted NLS-domain-dependent manner. Consequently, ROP16 knockout in PRU tachyzoites promoted the STING-mediated production of type I IFNs and restricted the replication of *T. gondii* during infection. In summary, we identified the function and molecular mechanism of ROP16 in regulating the cGAS-STING pathway for immune evasion. 

## 2. Materials and Methods

### 2.1. Cells and Parasite Culture

RAW264.7 cells (mouse macrophages), HEK293T cells (human embryonic kidney cells), and HFF cells (human foreskin fibroblasts) were obtained from the China Center for Type Culture Collection and maintained in our laboratory. RAW-Lucia ISG (mouse macrophages), RAW-Lucia ISGKO-cGas (*cGAS* knockout IRF-inducible Lucia luciferase reporter mouse macrophages), and RAW-Lucia ISG-KO-STING (*STING* knockout IRF-inducible Lucia luciferase reporter mouse macrophages) were gifts from Dr. Zhi-zhong Jing (Lanzhou Veterinary Research Institute, Chinese Academy of Agricultural Science, Lanzhou, China) and originally purchased from InvivoGen (InvivoGen, San Diego, CA, USA). The cells were cultured at 37 °C and 5% atmospheric CO_2_ in Dulbecco’s Modified Eagle’s Medium (DMEM, Thermo Fisher Scientific, Weltham, MA, USA) supplemented with 10% FCS (Thermo Fisher Scientific, Weltham, MA, USA), 2 mM L-glutamine (Thermo Fisher Scientific, Weltham, MA, USA), 100 U/mL penicillin, and 100 μg/mL streptomycin (Thermo Fisher Scientific, Weltham, MA, USA). The *T. gondii* PRU strain, a type II strain, was grown in HFF cells and used in this study.

### 2.2. Antibodies and Reagents

Anti-cGAS (#31659), anti-STING (#13647), anti-TBK1 (#3504), anti-IRF3 (#4302), anti-Phospho-STING (#72971), anti-Phospho-TBK1 (#5483), anti-Phospho-IRF3 (#4947) and anti-K36-linkage-specific polyubiquitin (#5621) were obtained from Cell Signaling Technology (Cell Signaling Technology, Danvers, MA, USA). Anti-GAPDH (#G8795), anti-FLAG (#F3165), HRP-Goat anti-Rabbit IgG (#A0545), and HRP-Goat anti-mouse (#A4416) were obtained from Sigma-Aldrich (Sigma-Aldrich, St. Louis, MO, USA). Anti-HA (#ab137838) was obtained from Abcam (Abcam, Cambridge, MA, USA). Alexa Fluor^TM^ 594 goat anti-rabbit IgG (#A11012) and Alexa Fluor^TM^ 488 goat anti-mouse IgG (#A11001) were obtained from Invitrogen (Invitrogen, Carlsbad, CA, USA). Protein A/G Plus-Agarose (#sc-2003) was purchased from SANTA CRUZ Biotechnology (SANTA CRUZ Biotechnology, Dallas, TX, USA). HiScript^®^ III RT SuperMix for qRT-PCR Kits (#R323) and AceQ Universal SYBR qRT-PCR Master Mix Kits (#Q511) were purchased from Vazyme (Vazyme, Nanjing, Jiangsu, China). TIANamp Genomic DNA Kit (#DP304) was purchased from TIANGEN (TIANGEN, Beijing, China). The primers used in this study are listed in Appendix A.

### 2.3. Plasmid Construction 

The full-length and truncated ROP16 were amplified via PCR using *T. gondii* PRU cDNA as a template and cloned into the pCDNA3.1-3×HA vector to yield the HA-tagged expression construct. Full-length cGAS, STING, TBK1, and IRF3 were amplified using mouse cDNA as a template and cloned into the pCDNA3.1-3×Flag vector to obtain Flag-tagged constructs. All the recombinant plasmids were analyzed and verified via DNA sequencing.

### 2.4. Transfection of RAW264.7 Cells

RAW264.7 cells were transfected using the Gene Pulser Xcell Electroporation System (Bio-Rad, Hercules, CA, USA) and according to the manufacturer’s protocol cultured at 37 °C under 5% atmospheric CO_2_ in DMEM supplemented with 2% FBS. 

### 2.5. Quantitative Real-Time Reverse Transcription PCR (qRT-PCR)

For the relative quantification assay, total RNA was extracted from infected or transfected cells using TRIzol reagent (Invitrogen, Carlsbad, CA, USA), and 1 μg of total RNA was reverse-transcribed using the HiScript^®^ III RT SuperMix for qRT-PCR Kit. The cDNA was quantified using the AceQ Universal SYBR qRT-PCR Master Mix Kit. The glyceraldehyde-3-phosphate dehydrogenase (GAPDH) gene was used as an internal control. The comparative threshold cycle (2^−ΔΔCt^) method was used to calculate the relative fold changes in mRNA expression [28]. 

For absolute quantification, the total genomic DNA was extracted from cells infected with *T. gondii* using the TIANamp Genomic DNA Kit. The TgB1 gene was engineered into pMD19-T and used as a standard. Different dilutions of pMD19-TgB1 were used to plot the standard curve. qRT-PCR was performed using one microgram of genomic DNA, and the copy number of TgB1 was calculated from the standard curve. The primers used for qRT-PCR are listed in Appendix A, and the standard curve is displayed in Appendix A.

### 2.6. Immunofluorescence Assay

The transfected cells were cultured on coverslips for 48 h, fixed with 4% paraformaldehyde for 20 min at room temperature, permeabilized with 0.1% Triton X-100 for 20 min at room temperature, blocked with 10% FBS in PBS for 1h at room temperature, probed with primary antibody at 4 °C overnight, and incubated in the dark with fluorescence-conjugated secondary antibody and Hoechst for 30 min at room temperature. Confocal fluorescence microscopy was performed on a Leica SP8 FALCON microscope (Leica Microsystems, Wetzlar, Germany) equipped with a Leica TCS SP8 X scanner.

### 2.7. Co-Immunoprecipitation 

RAW264.7 cells, co-transfected with plasmids encoding *T. gondii* ROP16 and host proteins, were harvested and lysed using 0.5 mL of RIPA Lysis Buffer (Beyotime, Haimen, Jiangsu, China) for 30 min on ice. The cell lysates were further ultrasonicated and clarified via centrifugation. For each immunoprecipitation experiment, 0.5 mL of the cell lysate was incubated with rabbit anti-HA antibody (1 μg) at 4 °C for 1 h on a 3D shaker. The antibody-bound proteins were further incubated with Protein A/G PLUS-Agarose (15 μL) at 4 °C overnight on a 3D shaker. The protein-bound beads were collected and washed six times with 0.75 mL of RIPA lysis Buffer. The final washed beads were resuspended in 50 μL of 1× sample buffer (GenScript, Nanjing, Jiangsu, China) and boiled for 10 min. 

### 2.8. Western Blotting Assay

The cell lysates and Co-IP samples were loaded onto 12% SDS-polyacrylamide gels for electrophoresis and transferred to a polyvinylidene fluoride (PVDF) membrane (EMD Millipore, Burlington, MA, USA). The membrane was blocked with 5% skim milk in Tris-buffered saline with Tween (TBST) at 4 °C overnight, incubated with the appropriate primary antibodies at 4 °C for 8–10 h, washed four times with TBST, incubated with HRP conjugated secondary antibodies at room temperature for 1 h, and washed six times with TBST. Protein bands were visualized using an enhanced chemiluminescence reaction buffer and imaged on a Tanon automatic chemiluminescence image analysis system according to the manufacturer’s protocol (Tanon, Shanghai, China).

### 2.9. Generation of ROP16 Knockout PRU (PRUΔROP16)

The *T. gondii* PRU ROP16 gene was knocked out using the CRISPR-Cas9 method according to a protocol reported by Shen Bang [29]. Briefly, Tachyzoites of the PRU strain were maintained in vitro in HFF monolayers. The freshly egressed tachyzoites were thoroughly released via syringe lysing with a 27-gauge needle, filtered through 3 µm polycarbonate membranes to eliminate the host cells, and washed thrice with sterile PBS. Their count was adjusted for subsequent experiments. A single-guide RNA (sgRNA) targeting the *ROP16* gene was generated and the pSAG1:CAS9-U6:sgUPRT plasmid was modified via PCR mutagenesis using specific primers (Appendix A). The DHFR+ resistance cassette was amplified simultaneously. *ROP16*-specific CRISPR-Cas9 plasmids and the DHFR+ resistance cassette amplicons were combined and co-transfected into *T. gondii* PRU tachyzoites using the Gene Pulser Xcell Electroporation System (Bio-Rad, Hercules, CA, USA). The transfected tachyzoites were cultured in HFF cells and selected five times using 3 μM pyrimethamine (Sigma-Aldrich, St. Louis, MO, USA) five times. Subsequently, clonal mutants were isolated and identified via PCR screening. The PRUΔROP16 tachyzoites were finally collected and used in further infection experiments.

### 2.10. T. gondii PRU Tachyzoites Infection In Vitro 

RAW264.7 cells, including RAW-Lucia ISG-KO-cGAS (rawl-kocgas), RAW-Lucia ISG-KO-STING (rawl-kostg) and RAW-Lucia ISG (rawl-isg), were, respectively, infected with PRU-WT or PRUΔROP16 tachyzoites at the dose of MOI = 1 or 2. At 2 h post-infection, the tachyzoites that failed to invade host cells were swilled out using sterile PBS, while those retained within host cells were cultured in DMEM. The immune responses of infected RAW264.7 cells were analyzed using Western blotting and qRT-PCR. The proliferation of *T. gondii* tachyzoites was measured using an immunofluorescence assay and qRT-PCR.

### 2.11. Statistical Analysis

All quantitative data are presented as the mean ± standard error of the mean (SEM) and were analyzed using GraphPad Prism (8.0, GraphPad Software Inc., San Diego, CA, USA). Statistical significance between the two groups was analyzed using unpaired Student’s *t*-test or one-way analysis of variance (ANOVA), using GraphPad Prism (8.0 GraphPad Software Inc., San Diego, CA, USA). At least three biological replicates were included.

## 3. Results

### 3.1. T. gondii ROP16 Is a Potential Inhibitor of the cGAS-STING Pathway 

The cGAS-STING axis plays a crucial role in the type I IFN response to *T. gondii* infection [20,21]. *T. gondii* can modulate type I IFNs signaling and induce immune evasion by injecting multitudes of proteins into the host cell [24,26]. Most of these proteins could interact with host immune systems by targeting different signaling pathways [6,11,26,30,31]. First, we investigated whether any *T. gondii*-secreted protein could target the cGAS-STING pathway. 

For this, we co-transfected RAW264.7 cells with plasmids encoding *T. gondii*-secreted protein (GRA3, ROP5, ROP16, GRA7, and ROP13) and a cGAS-encoding plasmid. cGAS-STING signaling inhibitors were screened based on the expression of IFN-β, IL-6, and CXCL10, as detected using RT-PCR. Of all the proteins, only TgROP16 failed to enhance the cGAS-STING-mediated transcription of IFNβ (Figure 1A). Since cGAS-STING signaling can also induce the expression of proinflammatory cytokines via the NF-κB and MAPK pathways, we quantified the expression of IL-6 and CXCL10. Consistently, ROP16 failed to enhance the transcription of IL-6 and CXCL10 in a cGAS-dependent manner (Figure 1B,C). These data indicate that ROP16 might inhibit cGAS-mediated-related immune responses.

### 3.2. T. gondii ROP16 Inhibited cGAS-STING-Mediated Signaling to Regulate Immune Responses

We further defined the involvement of the cGAS-STING pathway in ROP16-induced immune suppression. Signal transduction via this pathway is known to recruit TBK1 and trigger IRF3 phosphorylation [17]. Therefore, we used TBK1 and IRF3 phosphorylation as hallmarks of cGAS-mediated signaling and investigated the effects of ROP16 on these events by ectopically expressing ROP16- and cGAS-encoding plasmids in RAW264.7 cells. ROP16 significantly inhibited TBK1 and IRF3 phosphorylation in a dose-dependent manner, as evidenced by the results of Western blotting (Figure 2A–C). We also detected the expression of IFN-β, IL-12b, and CXCL10 in RAW264.7 cells using qRT-PCR. Because cGAS-STING signaling can be activated by any type of DNA, including ROP16-encoding plasmids, we defined the suppressive role of ROP16 by comparing the immune responses in cells transfected with high and low doses of ROP16-encoding plasmids. ROP16 suppressed the expression of proinflammatory cytokines in the high-dose group compared with the low-dose group (Figure 2D–F). To further confirm that this effect was mediated via the cGAS-STING pathway, we detected the function of ROP16 in cGAS- and STING-knockout RAW264.7 cells, respectively. qRT-PCR results showed that ROP16 failed to suppress the expression of proinflammatory cytokines in cGAS- or STING-knockout RAW264.7 cells in the high-dose group compared with the low-dose group (Figure 2H–J). It also failed to inhibit IRF3 and TBK1 phosphorylation in cGAS-knockout RAW264.7 cells in the high-dose group compared with the low-dose group, as evidenced by the results of Western blotting (Figure 2K–M). These results demonstrate that ROP16 could inhibit the expression of proinflammatory cytokines by targeting the cGAS-STING pathway.

### 3.3. T. gondii PRU ROP16-Deficiency Potentiates Innate Immune Responses 

To investigate the role of endogenous ROP16 during *T. gondii* infection, we generated ROP16-knockout PRU tachyzoites (PRUΔROP16) using the CRISPR-Cas9 method (Figure 3A, left). Compared with wild-type PRU, *ROP16* was significantly knocked out in PRUΔROP16, as evaluated using PCR (Figure 3A, right). We evaluated innate immune regulation by infecting RAW264.7 cells with PRU and PRUΔROP16. The qRT-PCR results demonstrated that IFN-β, IL-6, and ISG15 were transcribed significantly more in cells infected with PRUΔROP16 than in those infected with PRU (Figure 3B–D). Consistent with that, ROP16 knockout enhanced TBK1 and IRF3 phosphorylation by *T. gondii* in RAW264.7 cells (Appendix A, first three lines). 

To further validate the inhibition of host immune responses by ROP16 through the cGAS-STING pathway, we infected cGAS and STING-knockout RAW264.7 cells with PRUΔROP16 tachyzoites. qRT-PCR results indicated that PRUΔROP16 failed to upregulate IFN-β, IL-6, and ISG15 in cGAS-knockout cells (Figure 3E–G). Taken together, these results suggest that ROP16 played a critical role in inhibiting *T. gondii*-induced host immune responses via the cGAS-STING pathway. 

### 3.4. T. gondii ROP16 Interacted with STING 

To elucidate the mechanism through which ROP16 inhibited cGAS-mediated signaling, we attempted to identify the molecule it targeted. First, we co-overexpressed ROP16 with cCAS, STING, TBK1, and IRF3 in HEK293T cells and detected their subcellular localization using immunofluorescence. ROP16 was co-localized with cGAS, STING, TBK1, and IRF3 in the cytoplasm of host cells (Figure 4A), indicating that it may interact with the key components of cGAS-STING signaling. To further determine which protein interacted with ROP16, lysates from RAW264.7 cells co-transfected with ROP16- and host protein-encoding plasmids were analyzed using immunoprecipitation and Western blotting. We found that ROP16 was specifically associated with STING instead of cGAS, TBK1, or IRF3 (Figure 4B). To identify the ROP16 domains responsible for the ROP16-STING interaction, we engineered plasmids encoding full-length and truncated forms of ROP16 based on the domains predicted using online tools http://smart.embl-heidelberg.de/ (accessed on 3 August 2021) and Figure 4C and performed co-immunoprecipitation experiments with STING-encoding plasmids. We found that only the ROP16 truncation that retained the SignalP domain could interact with STING in a manner similar to the full-length protein (Figure 4D). Collectively, these results demonstrate that ROP16 was localized in the host cytoplasm, where it interacted with STING via its first N-terminal domain (SignalP). 

### 3.5. T. gondii ROP16 Inhibits cGAS-STING Signaling Based on the Predicted NLS Domain

To further characterize the ROP16 domain that played a key role in regulating the cGAS-STING signaling, full-length and truncated versions of ROP16 were co-expressed with cGAS in RAW264.7 cells. First, we detected TBK1 and IRF3 phosphorylation using Western blotting. The levels of p-TBK1 and p-IRF3 were higher in cells expressing ROP16-Δ1-344aa and ROP16Δ1-400aa, compared with those expressing full-length ROP16 (Figure 5A). This was confirmed via ImageJ densitometry analysis (Figure 5B,C). The transcription of the downstream signaling markers, such as IFN-β, IL-6, IL-16b, ISG15, and ISG56, was significantly increased in cells expressing ROP16-Δ1-344aa and ROP16Δ1-400aa, compared with those expressing full-length ROP16 (Figure 5D–H). These two loss-of-function ROP16 truncations lacked the NLS domain (Figure 4C). Taken together, these data indicate that the NLS domain of ROP16 played a crucial role in inhibiting cGAS–STING signaling. 

### 3.6. T. gondii ROP16 Inhibits the K63-Linked Polyubiquitylation of STING 

Having identified STING as the target of ROP16, we investigated how its activity was regulated by ROP16. Previous studies reported that the phosphorylation and K63-linked ubiquitination of STING are required for recruiting TBK1 and inducing IFN-Ⅰ. Therefore, we examined these active forms of STING after co-transfecting RAW264.7 cells with plasmids encoding recombinant cGAS and ROP16. Surprisingly, ROP16 failed to inhibit the phosphorylation of STING in a dose-dependent manner, as evidenced by Western blotting (Figure 6A,B). However, it inhibited the K63-linked ubiquitination of STING in a dose-dependent manner (Figure 6A,C). We also co-transfected RAW264.7 cells with plasmids encoding cGAS as well as full-length and truncated versions of ROP16 and found that the effects on K63-linked STING ubiquitination aligned with the results of the functional domain analysis performed previously (Figure 5 and Figure 6D,F), whereas the effects on STING phosphorylation did not (Figure 4 and Figure 6D,E). Taken together, these results indicate that ROP16 inhibited the STING activation by suppressing K63-linked ubiquitination rather than phosphorylation.

### 3.7. T. gondii ROP16 Plays an Important Role in T. gondii Immune Evasion

Since ROP16 could suppress the immune responses by inhibiting STING, we further explored its role in facilitating *T. gondii* immune evasion. Firstly, we infected the RAW264.7, RAW-KO-cGAS, and RAW-KO-STING cells with the PRU tachyzoites and detected the replication of *T. gondii* with the qRT-PCR method. The results showed that the replication of *T. gondii* PRU tachyzoites was significantly enhanced in cGAS- or STING-knockout RAW264.7 cells, compared with wild-type cells (Figure 7A). Moreover, it was approximately four-fold higher in STING-knockout cells than in cGAS-knockout cells (Figure 7A). These results reflect the crucial role played by cGAS-STING signaling in controlling *T. gondii* infection. Next, we found that *T. gondii* replication was inhibited in RAW264.7 cells infected with PRUΔROP16 tachyzoites, compared with those infected with PRU tachyzoites (Figure 7B), which indicates that ROP16 seriously affected *T. gondii* infection. To further determine whether ROP16 induced immune evasion dependent on the cGAS-STING pathway, RAW264.7 cells, and RAW-KO-STING cells were infected with PRU and PRUΔROP16 tachyzoites, respectively, and detected *T. gondii* loading by qRT-PCR. The results showed that PRUΔROP16 replication was attenuated compared with PRU replication in both RAW264.7 and RAW-KO-STING cells (Figure 7C,D). To further confirm the function of ROP16 in modulating PRU replication through the cGAS-STING pathway, immunofluorescence analysis was performed. Consistent with the load of PRU detected with the qRT-PCR method, the results showed that the PRU replicated faster in STING-deficient cells. ROP16-deficient PRU grew more slowly in both RAW264.7 and RAW-KO-STING cells than in wild-type PRU (Figure 7E,F). Taken together, these findings indicate that ROP16 likely induced immune evasion during *T. gondii* infection through the cGAS-STING pathway. 

## 4. Discussion

*T. gondii* infection occurs in two stages: an acute phase and a stable chronic phase [1,3]. The pathogen develops drug resistance when the infection progresses to the chronic phase, making the initial acute phase the most promising timeframe for controlling toxoplasmosis. The innate immune system plays a predominant role in host defense during the acute period. Understanding the mechanisms of *T. gondii*-induced innate immune evasion is important for developing prophylactic and therapeutic strategies for toxoplasmosis. 

In the past decade, many PRRs have been reported to sense *T. gondii*. Recently, several studies indicated that the cGAS-STING pathway is involved in controlling *T. gondii* infection [20,21]. cGAS, a newly identified cytoplasmic DNA sensor, has been well studied in infectious diseases, but its role during *T. gondii* infections is unknown. The source of the cGAS-STING pathway agonist (dsDNA) during *T. gondii* infection remains to be established. Based on the process of *T. gondii* infection [3,5], we speculated that the dsDNA may be derived in three ways: The genomic DNA of *T. gondii* may be divulged into the host cytoplasm during *T. gondii* replication; *T. gondii* may be injured by host immunity effectors, and DNA could be exposed to the host cytoplasm; and host cells may release their own DNA due to cellular damage. In other words, cGAS can be activated by DNA derived from both the host or *T. gondii*, and therefore the accurate agonist of the cGAS-STING pathway during *T. gondii* infection needs to be further identified. 

During infection, *T. gondii* secrets a multitude of proteins into the host cell to modulate host immune responses [6,11,25,27,30,32]. *T. gondii* GRA15 was found to enhance cGAS-STING signaling by potentiating STING activity during *T. gondii* infection [20]. ROP18 from the RH strain inhibited cGAS-STING signaling by targeting IRF3 to suppress type Ⅰ interferon [21]. The detailed mechanism through which *T. gondii* regulates type I IFN via the cGAS–STING pathway is still not clear. In the present study, we defined that ROP16 from the PRU strain could suppress cGAS-STING signaling and inhibit immune responses in mouse macrophages. Firstly, we found that ROP16 could inhibit cGAS-STING pathway in an ectopic expression system, as expected that ROP16 in high doses could suppress the phosphorylated of IRF3 and TBK1 (Figure 2A–C) as well as the expression of IFN-β, IL-12b, and CXCL10 (Figure 2D–F), in this system, the cGAS-STING signaling could be initialed by the transfected plasmid DNA. On the other hand, the expressed ROP16 had the inhibition function on cGAS-STING pathway. Here, we defined the suppressive role of ROP16 by focusing on the comparison of immune responses in high and low doses of ROP16-plasmid-transfected cells. Additionally, we infected RAW264.7 cells with ROP16-deficient PRU to further define the function of endogenous ROP16 in regulating cGAS-STING signaling, and we obtained similar results (Figure 3B–D). At the same time, this also indicates that cGAS-STING signaling could be initiated by *T. gondii* infection. Taken together, the results suggest that ectopically expressed and endogenous ROP16 play the same role in inhibiting the cGAS-STING signaling, independent of the activation forms of this pathway. 

To regulate cGAS-STING signaling, ROP16 needs to interact with the key composition of this pathway, so we screened the targeting factors with the immunofluorescence and co-immunoprecipitation analyses, and it was revealed that ROP16 interacted with STING, the adaptor for cGAS signaling, in the host cytoplasm (Figure 4). In addition, the subcellular location of the ectopically expressed ROP16 used in our study was not consistent with that of some nature forms reported previously (Figure 4A) [33]. Saeij et al. demonstrated that ROP16 was secreted into the nucleus, while it was mainly localized in the host cytoplasm in this study. The different expressing systems and protein transport systems may be the reason for these different results. However, these differences did not influence ROP16-STING interaction, and the function of recombined ROP16 is the same as natural proteins in regulating immune responses, which was further confirmed in our research on the ROP16-knockout PRU tachyzoites (PRUΔROP16) (Figure 3).

Furthermore, our findings reveal the binding mechanism and functional domains of ROP16. Co-immunoprecipitation analysis showed that only the ROP16 truncations that contained the SignalP domain could interact with STING (Figure 4D). We found that ROP16 mediated the interaction with STING based on the SignalP domain. The SignalP domain contains a predicted signal peptide, which is expected to determine the subcellular localization of ROP16. Furthermore, we found that ROP16 truncations containing the SignalP domain exhibited stronger inhibition of cGAS-STING signaling, while those lacking the NLS domain lost their inhibitory activity (Figure 5). We speculated that the NLS domain may play a critical role in inhibiting this pathway, while the SignalP domain may not be essential because lacking this domain (ROP16Δ1-23aa) had no effect on the inhibitory function of ROP16. The NLS domain may play a dual role: determining the subcellular localization and recruiting or activating DUBs to modulate STING signaling. Surprisingly, although the ROP16 truncations lacking the SignalP domain failed to bind to STING, they retained the ability to inhibit the cGAS-STING pathway. We speculate that ROP16 may inhibit STING through an indirect method. We found that ROP16 inhibited STING ubiquitination (Figure 6); however, ROP16 is a protein kinase, not a DUB, implying that it may regulate STING by recruiting DUBs instead of directly binding to it. 

We found that ROP16 inhibited cGAS-STING signaling by targeting STING, but the detailed mechanism of ROP16 regulating STING functions remains enigmatic. Previous studies indicated that STING phosphorylation at Ser365 was shown to control this pathway [34,35,36]. However, ROP16 did not inhibit STING phosphorylation at this site in a dose-dependent manner (Figure 6A,B,D,E). The K63-linked ubiquitination of STING also plays an important role in recruiting TBK1 and inducing type I IFN [37,38,39]. We found that ROP16 inhibited K63-linked STING ubiquitination in a dose-dependent manner (Figure 6A,C,D,F). Therefore, ROP16 inhibited cGAS-STING signaling by attenuating the K63-linked ubiquitination and not the phosphorylation of STING. However, as mentioned above, ROP16 is a PK instead of DUB, and thus the functions of ROP16 in inhibiting the ubiquitination of STING might be induced by recruiting other DUBs. The underlying molecular mechanism, including the identity of the recruited DUBs, remains to be elucidated.

We determined that ROP16 was crucial for *T. gondii* infection (Figure 7A), and the cGAS-STING pathway played a key role in limiting *T. gondii* infection (Figure 7B). However, ROP16 induced immune evasion even in cGAS- or STING-knockout cells (Figure 7C,D), implying that ROP16 could induce immune evasion via multiple pathways. Indeed, ROP16 has been shown to induce immune evasion via the STAT1-IRG axis [24], which is the effector phase of IFN-Ⅰ and IFN-Ⅱ, a *T. gondii*-induced immune response independent of the cGAS-STING pathway. These alternative pathways involving the IFN-STAT1-IRG axis also explain the recovery of PRUΔROP16 replication in STING-deficient cells.

*T. gondii* is a complex pathogen that elicits complicated immune responses. We found that the ROP16-induced inhibition of IFN-β, IL-6, and ISG15 was dependent on the cGAS-STING pathway (Figure 3E–G), but the ROP16-induced suppression of TBK1 and IRF3 phosphorylation was independent of it (Appendix A). Two reasons may explain the intricate function of ROP16. On the one hand, diverse PRRs, including TLR, RLR, and NLR, all of which transduce signal via phosphorylated TBK1 and IRF3, were involved in the immune response to *T. gondii* infection. ROP16 may suppress all these signaling pathways by inhibiting the phosphorylation of TBK1 and IRF3. On the other hand, although various pathways are involved in *T. gondii*-induced IFN-Ⅰ expression, cGAS-STING signaling may play a key role in mediating the ROP16-induced inhibition of IFN-Ⅰ responses.

In this study, ROP16 inhibited IL-12b expression. This contradicts previous research [33], wherein ROP16Ⅱ failed to inhibit IFN-γ or LPS-induced IL-12 expression. The main reason for this difference could lie in the initiating signal for IL-12. In that previous study, IL-12 was induced by IFN-γ or LPS, while we used transfected DNA for inducing IL-12 expression. These two studies focused on different regulatory pathways of IL-12 expression. The IFNγ-STAT3/6-IL12 pathway lies downstream of the cGAS-STING pathway and can be directly activated by IFNγ or LPS, independent of the cGAS-STING pathway. ROP16 can suppress IL-12 expression by targeting cGAS-STING signaling, as shown in this study. By contrast, in the previous research, the cGAS-STING pathway was bypassed, and hence, IL-12 expression was not inhibited. Additionally, ROP16 failed to inhibit the expression of proinflammatory cytokines (Figure 2H–J) in cGAS- or STING-deficient RAW264.7 cells, further confirming the finding that ROP16 inhibited immune responses by targeting the cGAS-STING pathway.

In summary, we found that *T. gondii* ROP16 suppressed innate immune responses by targeting the cGAS-STING pathway. Specifically, it inhibited the K63-linked ubiquitination of STING. This research highlights the novel role of *T. gondii* ROP16 in regulating anti-*T. gondii* responses within host cells and proposes a potential mechanism for *T. gondii* immune escape. These findings deepen our understanding of immune evasion by *T. gondii* and guide the development of strategies to prevent and control toxoplasmosis.

## Figures and Tables

**Figure 1 cells-12-01862-f001:**
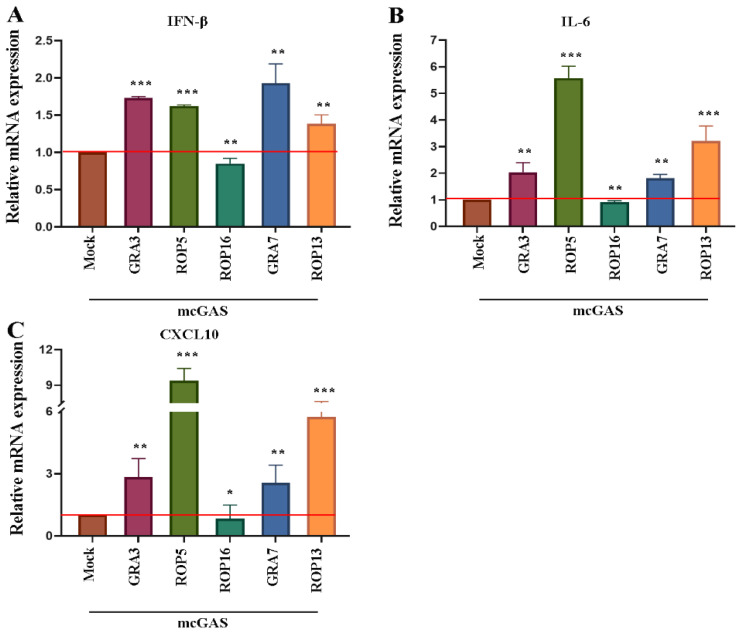
Screening for *T. gondii* proteins that regulate cGAS-STING mediated immune responses. RAW264.7 cells (1 × 10^6^ cells in 300 μL of opti-MEM) were co-transfected with cGAS (3 μg) and the indicated *T. gondii* protein-encoding plasmids (3 μg). After 24 h, the cells were harvested for qRT-PCR assays. The cGAS-dependent transcription of IFN-β (**A**), IL-6 (**B**), and CXCL10 (**C**) induced by *T. gondii* proteins was examined relative to that induced by empty vector transfection in RAW264.7 cells. The red line presented the average mRNA expression of MOCK group. Data are presented as the mean ± SEM of three independent experiments. Statistical analysis was performed using Student’s *t*-test. * 0.01 < *p* < 0.05; ** *p* < 0.01; and *** *p* < 0.001.

**Figure 2 cells-12-01862-f002:**
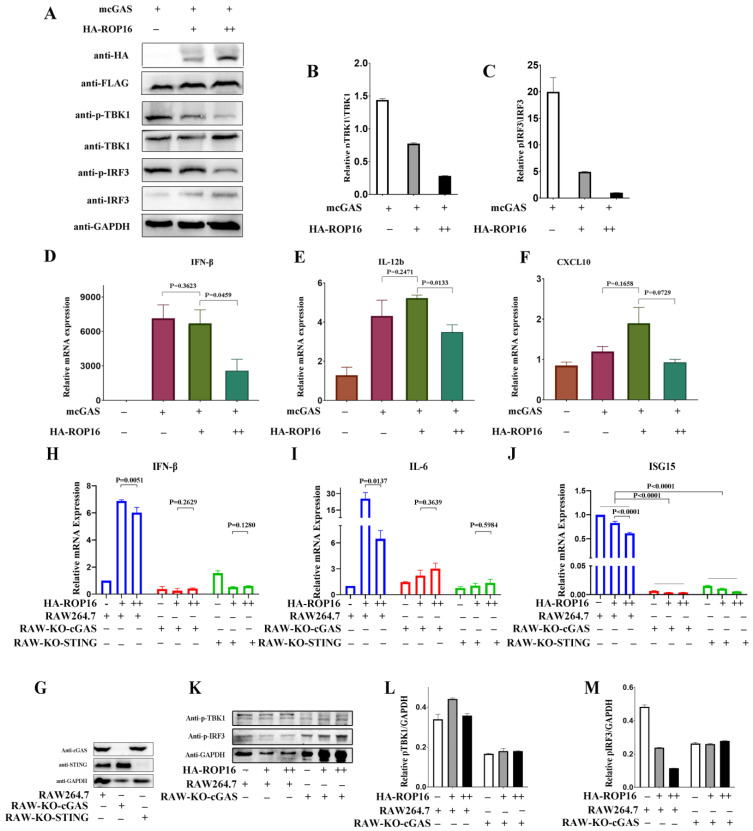
*T. gondii* ROP16 inhibits the activation of cGAS-STING signaling. RAW264.7 cells (1 × 10^6^ cells in 300 μL of opti-MEM) were co-transfected with cGAS (3 μg) and the indicated amounts of ROP16-encoding plasmids. Empty vectors were added to the cocktail to equalize the total amount of plasmid in each group. The cells were harvested for qRT-PCR and Western blotting 24 h and 48 h post-transfection, respectively: (**A**) the transfected RAW264.7 cells were lysed and analyzed using Western blotting; (**B**) ImageJ densitometry analysis was performed for pTBK1 relative to total TBK1 and (**C**) pIRF3 relative to total IRF3; (**D**–**F**) the transcription levels of IFN-β, IL-12b, and CXCL10 were quantified relative to that in empty vector-transfected RAW264.7 cells using qRT-PCR; (**G**) the expression of cGAS or STING was detected in RAW264.7, RAW-KO-cGAS, and RAW-KO-STING cells using Western blotting; (**H**–**J**) the transcription levels of IFN-β, IL-6, and ISG15 in RAW264.7, RAW-KO-cGAS, and RAW-KO-STING cells were quantified relative to that in empty vector-transfected RAW264.7 cells using qRT-PCR; (**K**–**M**) the activation of cGAS-STING signaling in RAW264.7 and RAW-KO-cGAS cells was detected using Western blotting. − un-added, + added, and ++ double. Data are presented as the mean ± SEM of three independent experiments. Statistical analysis was performed using Student’s *t*-test.

**Figure 3 cells-12-01862-f003:**
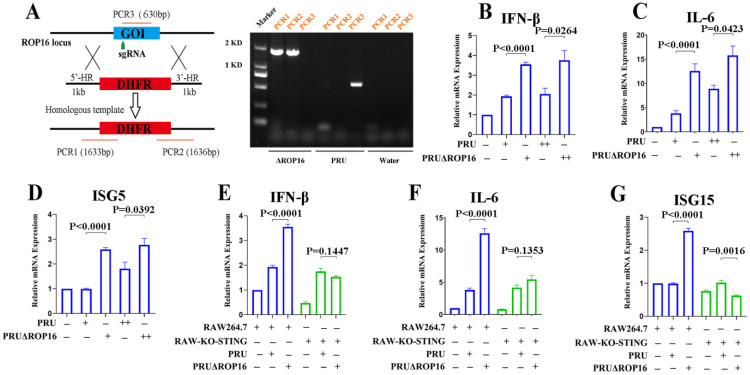
ROP16 knockout enhanced *T. gondii*-induced immune responses: (**A**)schematic diagram of CRISPR/CAS9-mediated DHFR insertion at the ROP16 locus (**Left**) and diagnostic PCRs on ROP16-knockout PRU tachyzoites (**Right**). PCR1–3 are products of diagnostic PCRs used to identify the PRU mutants. RAW264.7, RAW-KO-STING, and RAW-KO-cGAS cells were infected with PRU or PRUΔROP16 tachyzoites at different MOIs and harvested 8 h post-infection for qRT-PCR assays; (**B**–**D**) the mRNA expression of IFN-β, IL-6, and ISG15 was detected using qRT-PCR in RAW264.7 cells infected with PRU or PRUΔROP16 tachyzoites at different MOIs; (**E**–**G**) the mRNA expression of IFN-β, IL-6, and ISG15 was detected using qRT-PCR in RAW264.7 and RAW-KO-STING cells infected with PRU or PRUΔROP16 tachyzoites. − not added, + added, and ++ double. Data are presented as the mean ± SEM of three independent experiments. Statistical analysis was performed using Student’s *t*-test.

**Figure 4 cells-12-01862-f004:**
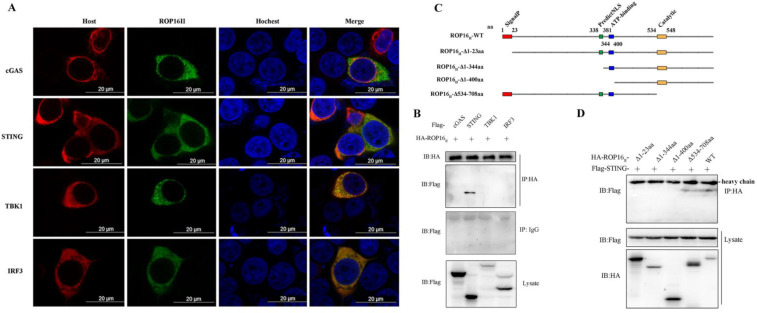
*T. gondii* ROP16 interacted with STING: (**A**) identifying the subcellular localization of ROP16 using immunofluorescence. HEK293T cells were co-transfected with plasmids encoding HA-ROP16 and Flag host proteins using Lip2000 reagents and subjected to immunofluorescence analysis 48 h post-transfection. HA-ROP16 was stained with Alexa Fluor^TM^ 488 goat anti-mouse IgG (green), while the indicated host proteins were stained with Alexa Fluor^TM^ 594 goat anti-rabbit IgG (red), and the nucleus was stained with Hoechst; (**B**) host proteins interacting with ROP16 were screened using Co-IP. RAW264.7 cells were co-transfected with plasmids encoding ROP16 and the indicated host proteins using the Gene Pulser Xcell Electroporation System, and they were harvested 48 h post-transfection for Co-IP and Western blotting using the indicated antibodies; (**C**) schematic representation of the mature form of *T. gondii* ROP16 and the truncated constructs used in this study. The signal peptide (red), putative nuclear localization sequences (PredictNLS, green), ATP-binding (blue) domain, and catalytic domain (orange) are represented; (**D**) screening of ROP16 domains that played key roles in binding to STING. RAW264.7 cells were co-transfected with plasmids encoding the indicated ROP16 truncations and STING using the Gene Pulser Xcell Electroporation System, and they were harvested 48 h post-transfection for Co-IP and Western blotting using the indicated antibodies. +: added.

**Figure 5 cells-12-01862-f005:**
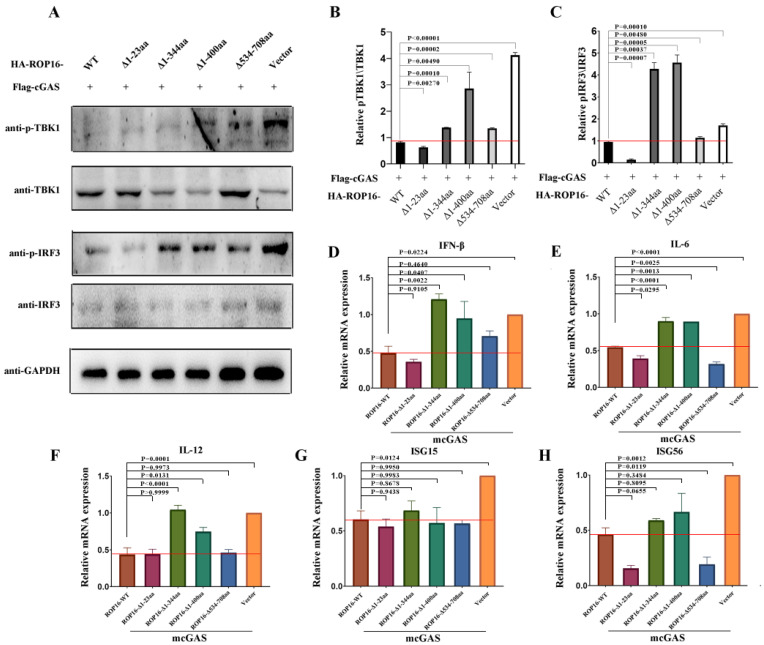
ROP16 inhibits cGAS-STING signaling dependent on the NLS domain. RAW264.7 cells were co-transfected with plasmids encoding truncated versions of ROP16 and host cGAS using the Gene Pulser Xcell Electroporation System, and they were harvested 24 h and 48 h post-transfection for qRT-PCR and Western blotting with the indicated antibodies, respectively; (**A**) transfected RAW264.7 cells were lysed and indicated proteins were analyzed using Western blotting; (**B**) ImageJ densitometry analysis for p-TBK1 relative to total TBK1 and (**C**) p-IRF3 relative to total IRF3; (**D**–**H**) the transcription levels of IFN-β, IL-6, IL-12b, ISG15, and ISG56 were quantified relative to that in RAW264.7 cells transfected with empty vector. Data are presented as the mean ± SEM of three independent experiments. The red line presented the average mRNA expression of Wild-type of ROP16 protein expressed group. + added. Statistical analysis was performed using Student’s *t*-test.

**Figure 6 cells-12-01862-f006:**
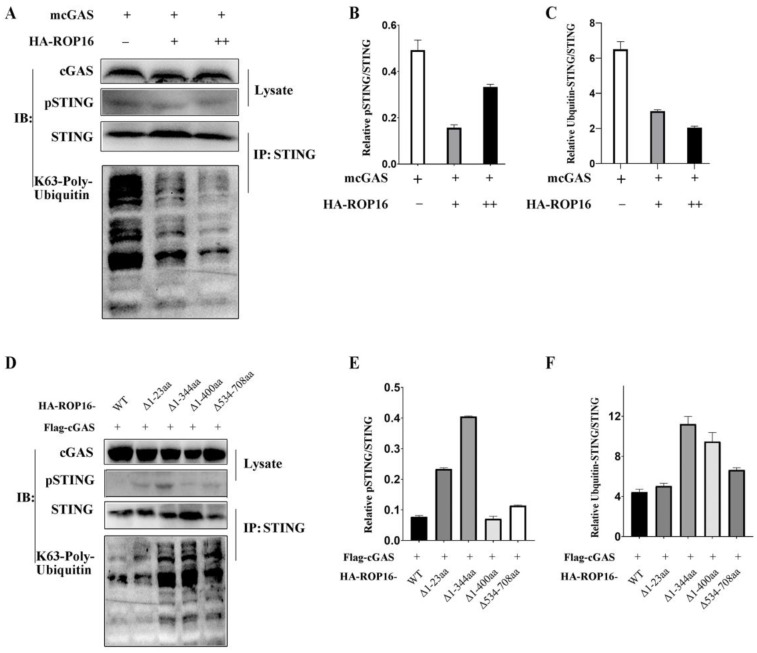
ROP16 inhibits the polyubiquitination of STING. RAW264.7 cells were co-transfected with plasmids encoding truncated versions of ROP16 and host cGAS using the Gene Pulser Xcell Electroporation System, and they were harvested 48 h post-transfection for IP and Western blotting: (**A**,**D**) cell lysates were subjected to Western blotting with anti-cGAS and anti-Phospho-STING antibodies (**Top**). Equal amounts of lysate were first immunoprecipitated with an anti-STING antibody and finally detected using Western blotting with an anti-K63-linkage-specific polyubiquitin antibody (**Bottom**); (**B**,**E**) ImageJ densitometry analysis for pSTING relative to total STING, and (**C**,**F**) for K63-linkage-specific polyubiquitin–STING relative to total STING. − not added, + added, and ++ double. Data are presented as the mean ± SEM of three independent experiments.

**Figure 7 cells-12-01862-f007:**
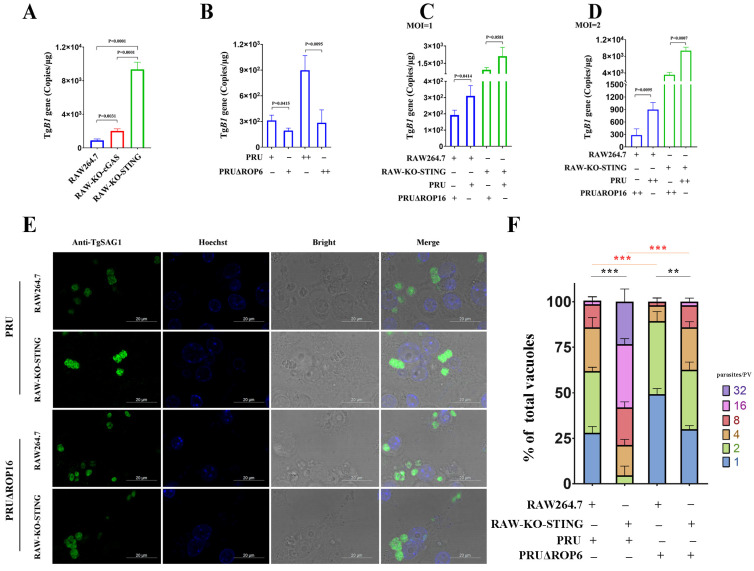
ROP16 is important for *T. gondii* immune evasion. RAW264.7, RAW-KO-cGAS, and RAW-KO-STING cells were infected with PRU or PRUΔROP16 tachyzoites at different MOIs and harvested 96 h post-infection to determine the copies of TgB1 by qRT-PCR: (**A**) qRT-PCR was used to detect TgB1 in PRU-infected RAW264.7, RAW-KO-cGAS, and RAW-KO-STING cells; (**B**) qRT-PCR was used to detect TgB1 in RAW264.7 cells infected with PRU or PRUΔROP16 tachyzoites; (**C**,**D**) qRT-PCR was used to detect TgB1 in RAW 264.7 and RAW-KO-STING cells infected with PRU or PRUΔROP16 tachyzoites at different MOIs; (**E**) RAW264.7, RAW-KO-cGAS, and RAW-KO-STING cells were cultured on coverslips and infected with PRU or PRUΔROP16 tachyzoites, respectively. At 48 hpi, tachyzoites replication was detected with an assay using IFA; (**F**) the number of parasites in each parasitophorous vacuole (PV) was valued based on the IFA results. − not added, + added, and ++ double. Data are presented as the mean ± SEM of three independent experiments. Statistical analysis was performed using Student’s *t*-test. ** *p* < 0.01; and *** *p* < 0.001.

## Data Availability

The authors confirm that the data supporting the findings of this study are available within the article.

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
