# Peer review of "ROP16 of Toxoplasma gondii Inhibits Innate Immunity by Triggering cGAS-STING Pathway Inactivity through the Polyubiquitination of STING"

_cells, 2023, doi:10.3390/cells12141862_

Round 1

Reviewer 1 Report

Why don't the authors explain how they are isolating the tachyzoites for transfection, considering that the parasite is intracellular? How do they distinguish between host cells and parasites? How many hours can the parasite survive outside host cells?

In my opinion, in order to quantify the number of replicating parasites, real-time quantitative PCR (qPCR) should be performed together with staining of infected cells (such as Giemsa staining). Additionally, the authors should present the standard curve for real-time PCR.

Reviewer 2 Report

1. PAGE 1 – line 26 – “… limits the replication of T. gondii.

T. gondii in italics.

2. PAGE 1 – line 44 -  “…gondii by sensing distinct sets of PAMP abovementioned.. Replace  abovementioned  for  above mentioned

3. PAGE 2 – LINES 92-95.  “The RAW264.7 cells, HEK293T cells and HFF cells were obtained from China Center for Type Culture Collection and maintained in our laboratory”. RAW-Lucia ISG (rawl-isg), RAW-Lucia ISGKO-cGas (rawl-kocgas) and RAW-Lucia ISG-KO-Sting (rawl-kostg) were originally purchased from the InvivoGen Company (San Diego, California, USA). 

Suggestion: Inform the cell type and origin of these cell lines.

4. PAGE 6 – line 237:  “The Western Blotting results  showed similar results, as shown in fig2K-M... “

Suggestion: Avoid duplication of words in the same sentence.

5. Page 7 – lines 260-262: “ For further defining the function of ROP16 in inhibiting immune responses during T. gondii infection, we investigated the effect of endogenous ROP16 on regulating immune  response during T. gondii infection.”

Suggestion: Rewrite this sentence.

The manuscript aimed to investigate the function of ROP16 (one of the proteins secreted by T. gondii rhoptrias during its intracellular development) and its mechanism that would regulate the cGAS-STING pathway, which induces the evasion of the parasite from the host's immune system. cGAS-STING signaling is an important pathway for inducing host defense against T. gondii infection, therefore a target of great interest in the context of the pathology generated by the parasite. Despite its importance, understanding the strategies that the parasite develops to neutralize the host's immune system has aroused the interest of different groups, which fully justifies the development of this research.

Thus, the authors of this article conducted the research rationally, in a line of molecular and immunological investigation targeting the ROP16 protein and tracked the cGAS-STING signaling and identified that this protein is capable of inhibiting this pathway and consequently regulating the evasion of the parasite of the host's immune system.

I consider the article to have an impact in the context of understanding the biology of T. gondii, contributing to clarifying the mechanisms generated by the parasite to ensure the success of its parasitism.

Moderate editing of English language required.

Round 2

Reviewer 1 Report

Thanks to the authors to accept my suggestions, good work